# Sustained Elevated Circulating Activin A Impairs Global Longitudinal Strain in Pregnant Rats: A Potential Mechanism for Preeclampsia-Related Cardiac Dysfunction

**DOI:** 10.3390/cells11040742

**Published:** 2022-02-21

**Authors:** Bhavisha A. Bakrania, Ana C. Palei, Umesh Bhattarai, Yingjie Chen, Joey P. Granger, Sajid Shahul

**Affiliations:** 1UQ Centre for Clinical Research, Faculty of Medicine, University of Queensland, Brisbane 4029, Australia; 2Department of Physiology and Biophysics, University of Mississippi Medical Center, Jackson, MS 39216, USA; ubhattarai@umc.edu (U.B.); ychen2@umc.edu (Y.C.); jgranger@umc.edu (J.P.G.); 3Department of Surgery, University of Mississippi Medical Center, Jackson, MS 39216, USA; apalei@umc.edu; 4Department of Anesthesia and Critical Care, University of Chicago, Chicago, IL 60637, USA; sshahul1@dacc.uchicago.edu

**Keywords:** preeclampsia, cardiac dysfunction, placental factors, activin A

## Abstract

Mediators of cardiac injury in preeclampsia are not well understood. Preeclamptic women have decreased cardiac global longitudinal strain (GLS), a sensitive measure of systolic function that indicates fibrosis and tissue injury. GLS is worse in preeclampsia compared to gestational hypertension, despite comparable blood pressure, suggesting that placental factors may be involved. We previously showed that Activin A, a pro-fibrotic factor produced in excess by the placenta in preeclampsia, predicts impaired GLS postpartum. Here, we hypothesized that chronic excess levels of Activin A during pregnancy induces cardiac dysfunction. Rats were assigned to sham or activin A infusion (1.25–6 µg/day) on a gestational day (GD) 14 (*n* = 6–10/group). All animals underwent blood pressure measurement and comprehensive echocardiography followed by euthanasia and the collection of tissue samples on GD 19. Increased circulating activin A (sham: 0.59 ± 0.05 ng/mL, 6 µg/day: 2.8 ± 0.41 ng/mL, *p* < 0.01) was associated with impaired GLS (Sham: −22.1 ± 0.8%, 6 µg/day: −14.7 ± 1.14%, *p* < 0.01). Activin A infusion (6 µg/day) increased beta-myosin heavy chain expression in heart tissue, indicating cardiac injury. In summary, our findings indicate that increasing levels of activin A during pregnancy induces cardiac dysfunction and supports the concept that activin A may serve as a possible mediator of PE-induced cardiac dysfunction.

## 1. Introduction

During normal gestation, the woman’s cardiovascular system endures major structural and hemodynamic alterations to meet the demands of the developing fetus. The pregnancy-induced hemodynamic shift usually initiates prior to placentation, peaks in the second trimester of gestation, and persists until delivery [1]. However, disturbed cardiovascular adaptations may lead to adverse pregnancy outcomes, including peripartum cardiomyopathy and preeclampsia (PE). PE is a syndrome characterized by new-onset hypertension and significant end-organ damage in the last half of gestation or immediate postpartum period [2]. It affects 3–5% of pregnancies worldwide, with incidence increasing over the recent decades due to the higher prevalence of risk factors such as advanced maternal age, obesity, and other chronic health conditions [3,4,5]. In addition to low plasma volume, high peripheral vascular resistance, and reduced cardiac output [6,7,8], the heart of PE women often exhibits impaired global longitudinal strain (GLS) [9,10,11,12,13,14,15]–a sensitive measure of systolic function that indicates cardiac injury and fibrosis. Although hypertension and cardiac dysfunction generally resolve after delivery, PE is associated with a significant risk of mortality as well as short- and long-term morbidity for both mother mothers and offspring [16]. Despite the negative clinical, social, and public health impact of PE, the only definitive treatment is early delivery [17]. Therefore, it is imperative to determine the mechanisms underlying cardiovascular dysfunction during pregnancies complicated by PE in order to advance the discovery of novel therapies to improve maternal and fetal outcomes.

The cardiovascular adaptations of pregnancy are mainly regulated by hormonal and neural mechanisms, with multiple pathways interacting to control blood pressure [18]. However, as an initial step in the pathophysiology of PE, abnormal placental morphogenesis and perfusion stimulate the release of placental factors into the maternal circulation, and compelling evidence indicates that vascular and cardiac dysfunction in PE are also induced by many of these placental factors [17]. Indeed, it was found that GLS gradually deteriorates from normotensive pregnancy to gestational hypertension (GH, allegedly without placental compromise) to PE [9,14,15], despite comparable blood pressure in these hypertensive groups [14]. Additionally, GLS remains impaired postpartum even after adjustment for blood pressure and other clinically and biologically relevant variables [19,20,21], further supporting a role for placental factors in the development of cardiac dysfunction in PE.

One potential placental factor that may contribute to detrimental changes to the heart in PE is activin A [22]. Activin A is a glycoprotein member of the transforming growth factor β superfamily involved in multiple biological functions, including reproduction, embryogenesis, inflammation, and fibrosis. While the placenta is the major source of activin A in the maternal circulation, the pituitary gland, ovaries, uterus, and inflammatory cells are also able to produce activin A [23]. Increased expression of activin A was reported in patients with heart failure as well as in experimental animal models of myocardium infarction and dilated cardiomyopathy [24,25,26]. Furthermore, activin A was implicated in cardiac remodeling and fibrosis by promoting the release of atrial natriuretic peptide (ANP), brain natriuretic peptide (BNP), reactive oxygen species, and cytokines from cardiac myocytes in vitro [27,28].

Dysregulation of the activin A signaling pathway is linked to the manifestation and prediction of PE. Activin A is increased in the blood and placental tissue of PE patients compared with both GH and normal pregnant women [29,30,31,32,33,34,35,36,37,38,39,40,41,42,43]. Moreover, circulating levels of activin A are higher from 10 to 26 weeks of gestation in women who subsequently developed PE [33,35,44,45,46,47,48,49,50]. We previously showed that serum activin A levels during the third trimester in normotensive pregnancy, gestational or chronic hypertension, and PE correlate positively with abnormal GLS at one year postpartum [21] and this relationship persists 10 years after pregnancy [20]. Earlier studies demonstrated sustained infusion of activin A-induced PE-like features in pregnant mice, including hypertension, proteinuria, preterm birth, and fetal growth restriction [51]. However, the importance of activin A in mediating cardiac dysfunction related to PE is not known. Thus, the present study aimed to test the hypothesis that chronic excess levels of activin A induce cardiac dysfunction in pregnant rats.

## 2. Materials and Methods

### 2.1. Experimental Animals and Protocols

All experiments were approved by the Institutional Animal Care and Use Committee at the University of Mississippi Medical Center (UMMC) and conducted in accordance with the National Institutes of Health (NIH) Guide for the Care and Use of Laboratory Animals. Female Sprague Dawley rats were received from Charles River Laboratories (Wilmington, MA, USA) on gestational day (GD) 10 and housed in the Center for Comparative Research facility at the UMMC. On GD 14, animals were randomly assigned to either the sham (*n* = 10) or the recombinant activin A (AnshLabs, Webster, TX, USA) group. For activin A infusion, rats were anesthetized (~3% isoflurane in 2 L/min O_2_), and mini-osmotic pumps (Alzet, Cupertino, CA, USA) containing varying doses were placed intraperitoneally. Pumps were prepared to infuse 1.25 µg/day, 1.9 µg/day, 3 µg/day, or 6 µg/day of activin A continuously from GD 14 to 19 (*n* = 6 per dose). These doses were selected to achieve a five-fold increase of Activin A in pregnant rats, to mimic levels seen in PE, compared to healthy pregnancies, clinically [21,40]. All animals were fed nutritionally complete diets (Envigo Teklad 8640, Indianapolis, IN, USA) and water, ad libitum.

On GD 19, blood pressure was measured, followed by an echocardiogram and sacrifice. These procedures are detailed below. During sacrifice, blood was collected from the abdominal aorta into EDTA-coated vacutainers (BD, Franklin Lakes, NJ, USA). The heart was excised, arrested in ice-cold cardioplegic solution, and weighed. The coronary circulation was cleared with a cardioplegic solution by retrograde perfusion, snap-frozen, and stored at −80 °C. To calculate fetal outcomes, pup and placental weights were averaged per animal to constitute a single data point.

### 2.2. Maternal Blood Pressure Measurement

Blood pressure was assessed as we previously described [52]. Briefly, on GD 18, indwelling carotid catheters were placed to measure blood pressure via a pressure transducer. On GD 19, conscious blood pressure was measured in all animals for a period of two hours. The final 30 min was used to average mean arterial pressure (MAP).

### 2.3. Echocardiography with Speckle Tracking Technology

Comprehensive echocardiography was performed using the Vevo3100 (VisualSonics, Toronto, ON, Canada) and an MX250 scan head for small animals on GD 19. Rats were anesthetized with constant temperature and heart rate monitoring. Cardiac output and ejection fraction were determined in B-mode, whereas mass was determined in M-mode. Speckle tracking analysis to determine GLS was performed using the accompanying VisualSonics VevoStrain software.

### 2.4. Western Blotting

Quantification of the β-myosin heavy chain (β-MHC) was performed by Western blotting as we previously described [53,54]. Briefly, rat heart samples were homogenized in RIPA lysis buffer (Thermo Scientific, Rockford, IL, USA) containing protease and phosphatase inhibitors. Protein concentration in lysed samples was estimated using the Pierce BCA protein assay kit (Thermo Scientific). Samples were loaded and separated by SDS-PAGE gel and transferred to nitrocellulose membrane. The membrane was then blocked with 5% non-fat milk in TBST for 1 h and incubated with primary antibodies overnight at 4 °C. The antibodies used were mouse anti-β-MHC (cat# ab50967, Abcam, Waltham, MA, USA ) or mouse anti-GAPDH (cat# 97166S, Cell Signaling Technology, Inc., Danvers, MA, USA). The membrane was washed with TBST three times, 10 min each, and then incubated with hrp-conjugated donkey anti-mouse secondary antibody for 1 h. The protein bands were detected using the iBright FL1500 instrument (Thermo Scientific), and the bands were quantified using NIH ImageJ analysis software (Version: 1.52k, Bethesda, MD, USA).

### 2.5. Histology for Detection of Cardiac Fibrosis

Fibrosis was determined by cardiac histology as we previously described [53,55]. Briefly, heart samples were collected, fixed in 10% formalin, and embedded in paraffin. The paraffin blocks were sectioned into 10 µm sections and stained with Sirius Red and the Fast Green Collagen Staining Kit (Chondrex Inc., Woodinville, WA, USA), according to the manufacturer’s instructions. The slides were then washed and mounted with a mounting medium. Images were taken with a bright field microscope and analyzed using NIH ImageJ software.

### 2.6. Biochemical Analyses

Following sacrifice on GD 19, blood was spun at 2500 RPM for 12 min at 4 °C and stored at −20 °C. Activin A (AnshLabs), ANP, and BNP (ab108797 and ab108815, respectively, Abcam) were measured in serum using commercially available sandwich ELISA kits, as per the manufacturers’ instructions. ANP and BNP levels were also quantified in heart homogenate samples by ELISA (Abcam) and normalized by total protein concentration determined by BCA assay (Thermo Scientific). A Tecan GENios plate reader (Mannedorf, Switzerland) with Magellan version 4.1 software (Mannedorf, Switzerland) was used to read the plates.

### 2.7. Statistical Analysis

All graphs and statistical analysis were performed using Graphpad Prism (Version 9.0.0 GraphPad Software, La Jolla, CA, USA). The specific n for each data set is detailed in the figure legend. Data followed a normal distribution and were analyzed using either a two-tailed t-test or one-way ANOVA followed by Tukey’s multiple comparison test depending on the number of groups being compared. Pearson’s correlation was performed to determine the relationship between circulating activin A and markers of cardiac injury. Data are presented as mean ± SEM. A *p*-value of 0.05 was considered statistically significant.

## 3. Results

Activin A-infused pregnant rats had circulating activin-A levels on GD 19 that increased in a dose-dependent manner. We trialed four different doses, and at 6 µg/day, activin A concentration in serum was approximately five times that of the sham group (Figure 1), which is comparable to the fold-change increase seen in PE women [21,40]. GLS, which is a sensitive measure of cardiac systolic function, was significantly decreased in pregnant rats treated with 1.25, 1.9, 3, or 6 µg per day of activin A compared to sham (Figure 2). Interestingly, MAP was not different among groups regardless of the activin A dose infused (Figure 3), suggesting that the effects of activin A on GLS were independent of blood pressure.

Pregnant rats infused with the highest dose of activin A (6 µg/day) did not have any significant differences in fetal parameters on GD 19, including fetal weight and placental weight (Table 1), compared to sham rats. Similarly, maternal body weight, heart weight, heart rate, cardiac output, ejection fraction, and fractional shortening were not significantly altered in activin A-infused pregnant rats (Table 1, Figure 4).

Previous studies demonstrated that pathological left ventricular hypertrophy is often associated with abnormal accumulation of collagen within the heart extracellular space, with the resultant fibrosis leading to increased ventricular stiffness. Conversely, perivascular fibrosis may cause myocardial ischemia. Both ventricular stiffness and myocardial ischemia were demonstrated to play important roles in the development of cardiac dysfunction [56]. Furthermore, in a variety of pathophysiologic conditions, including hypertrophy and ischemia, the postnatal heart undergoes adaptive mechanisms to support cardiac structure and function, such as the switch from α-MHC to β-MHC. However, at a certain point, this fetal-like reprogramming no longer suffices, with increased β-MHC linked to heart failure [57]. To characterize whether circulating activin A in excess induces cardiac fibrosis during pregnancy, collagen I and III fibers were visualized and quantified in histological heart slides using Sirus red staining. We also determined the fetal gene program by quantifying β-MHC in heart homogenates using Western blotting. While there were no significant differences between sham and activin A (6 µg/day)-infused pregnant rats in left ventricular fibrosis (Figure 5A,B), β-MHC protein expression was significantly increased in the left ventricle of activin A-infused rats on GD 19 (Figure 5C,D).

Finally, we measured ANP and BNP in the circulation and heart tissue of sham rats and rats that were infused with 6 µg/day of activin A on GD 19 as indicators of presence and severity of hemodynamic cardiac stress and heart failure [58]. There were no statistically significant differences in either serum or left ventricular ANP and BNP levels between groups (Figure 6A–D). Due to limitations on the standard curve of the ELISA kit, six (out of 10) serum ANP values in the sham group were undetectable compared with one (out of six) value in the activin A-infused group (Figure 6A). Nonetheless, circulating BNP levels were measurable in all serum samples from both groups (Figure 6B). Unfortunately, we had no remaining heart samples from two of the activin A-infused pregnant rats to run the ANP and BNP assays (Figure 6C,D). Interestingly, circulating activin A was positively correlated to circulating BNP, but a significant relationship was not detected for circulating ANP, suggesting that activin A induced some degree of cardiac injury (Figure 7A,B).

## 4. Discussion

This study sought to determine whether increases in circulating activin A induces cardiac dysfunction during late pregnancy and whether this was associated with fibrosis and markers of cardiac injury. We previously showed that a five-fold increase in plasma activin A levels of PE patients, compared to normal pregnant women, predicts cardiac dysfunction at 1 year postpartum [21]. Similarly, we demonstrated here that a five-fold increase in circulating activin A levels induces cardiac dysfunction, as assessed by GLS, in pregnant rats. While fibrosis was not detected in the left ventricle of activin A-infused rats, we report a pronounced activation of the fetal gene program, which is indicative of cardiac injury. These data suggest that elevated circulating activin A in PE may play a role in mediating cardiac dysfunction in this cohort.

### 4.1. Cardiac Dysfunction in Preeclampsia Is Associated with Elevated Activin A

Circulating placental factors play a central role in promoting end-organ damage in PE. For instance, maternal angiogenesis imbalance, including increased soluble fms-like tyrosine kinase (sFlt)-1 and soluble endoglin (sEng) as well as decreased placental growth factor (PlGF), were associated with cardiac abnormalities during gestation and postpartum [59,60]. More recently, elevated antepartum activin A was associated with impaired GLS during gestation and also with worsened GLS one year postpartum. At the plasma activin A levels above 23.74 ng/mL, 84.7% of patients developed impaired GLS postpartum compared to 24.5% of those below the cutoff value [21]. These associations between circulating placental-derived factors and cardiac dysfunction remain significant after multivariable adjustment for clinically relevant confounders, including blood pressure [21,59,60]. Importantly, while circulating levels of other placental factors are similar between normal pregnancy and PE [59], activin A levels are comparable at one year postpartum but significantly elevated at approximately 10 years after a pregnancy complicated by PE [20,21]. These clinical data provide strong evidence of a relationship between plasma activin A and a lifelong risk of cardiac dysfunction. However, the direct effect of excess circulating activin A on heart function during pregnancy has yet to be examined.

The primary source of activin A during preeclampsia is thought to be the placenta; indeed, placental activin A expression positively correlates with circulating activin A levels in PE patients [29,31,37,41]. However, in vitro studies show that trophoblasts as well as endothelial and immune cells also secret activin A when exposed to pro-inflammatory and stress oxidative conditions, both hallmarks of PE [23]. Although the source of elevated circulating activin A levels in the years following the index pregnancy is not known, it is plausible that these other cell types may contribute.

Prior studies showed that activin A-infused pregnant mice exhibit increased blood pressure levels compared with saline-infused counterparts, along with preterm birth and fetal growth restriction [51]. However, we noted that impaired GLS following activin A infusion in pregnant rats was independent of alterations in blood pressure. Moreover, there were no differences in gestational length and fetal weight in our study. Differences in the effect of activin A on maternal-fetal outcomes between studies might be related to the activin A administration regimen, methods employed for blood pressure measurement, and/or even inter-species variability. Lim et al. infused activin A subcutaneously at a dose of 360 µg/kg/day from GD 10 to 16, resulting in more than an eight-fold increase in serum activin A levels (216 ± 102 vs. 1845 ± 286 ng/mL). Additionally, they assessed maternal blood pressure by noninvasive tail-cuff [51]. In contrast, we invasively determined a method considered more accurate and reliable than plethysmography via a catheter implanted in the carotid [61]. Indeed, the results of our study isolate the role of elevated circulating activin A levels in the cardiac function of pregnant rats, independent of a PE-like phenotype or hypertension itself.

### 4.2. Mechanisms of Activin A-Induced Cardiac Dysfunction

The mechanisms by which activin A results in cardiac dysfunction were not previously studied in the context of pregnancy. In mouse models of aging and left ventricular pressure overload, circulating activin A levels and cardiac activin type II receptor (ActIIR) signaling is significantly increased [62]. Interestingly, cardiac activin A expression was similar in aged (28 months old) animals compared to young animals (4 months old), suggesting that elevated activin A in the circulation and consequent activation of the ActIIR signaling in heart tissue largely originates from outside the heart. However, cardiac activin A expression also increased after pressure overload in young animals, indicating that both local and systemic activin A production were up-regulated [62]. Furthermore, overexpression of activin A with the administration of an adenoviral construct into young mice was sufficient to impair radial systolic and early diastolic strain rates without significantly altering blood pressure in only 96 h after injection [62]. Similarly, we report here that five days of continuous infusion of activin A at 1.25, 1.9, 3, or 6 µg/day result in significant decreases in GLS with no effect on blood pressure. These data are consistent with clinical data and suggest that the activin A-ActIIR pathway plays a causal role in cardiac dysfunction.

A number of studies have implicated activin A overexpression in the development of fibrosis [63,64]. While the impaired systolic strain is linked to increased fibrosis [65], few studies investigated the direct relationship between activin A overexpression and the development of cardiac fibrosis, and whether this effect could be ameliorated by inhibition of activin A receptors. Hu et al. showed that activin A stimulates cardiac fibroblast proliferation and differentiation [66]. Roh et al. demonstrated that a monoclonal antibody that blocks ActIIR named CDD866 (a murinized version of bimagrumab) ameliorates subclinical left ventricular systolic function of old mice. Moreover, CDD866 was not only able to mitigate substantially the decline in systolic function induced by pressure overload but also to attenuate the expression of fibrosis-related genes in heart tissue, both in prevention and treatment protocol [62]. The same research group found that cardiac myocyte-specific ActIIR knockout mice display normal cardiac structure and function at baseline but were protected from pressure overload-induced systolic dysfunction [62]. Castillero et al. also showed that inhibition of the ActIIR signaling either with decoy myostatin to prevent ligands from binding to ActIIR or with follistatin was associated with preserved cardiac function and fibroblast-driven decrease in cardiac fibrosis in mice following coronary ligation to induce myocardial infarction [63]. Importantly, treatment with these ActIIR inhibitors was first administered two weeks after the initial insult [62,63]. As mentioned previously, we continuously infused activin A for only 5 days. Thus, the infusion window may have been too short to detect significant differences in cardiac fibrosis between groups of pregnant rats.

### 4.3. The Relationship between Activin A and Recognized Markers of Cardiac Dysfunction

Elevated β-MHC protein expression indicates the activation of the fetal gene program, which occurs in response to stress. The predominant sarcomeric proteins in the fetal heart switch from β-MHC to α-MHC in rodents to support the mechanical performance and efficiency of the heart ex utero [57]. Abnormal expression of these MHC isoforms was reported in cardiac hypertrophy and heart failure. Although β-MHC presents lower adenosine triphosphatase activity and lower filament sliding velocity, it can generate cross-bridge force with a higher economy of energy consumption than α-MHC, suggesting that a shift from α- to β-MHC may be an adaptative response in order to preserve energy. Furthermore, culminating evidence indicates that increased β-MHC expression decreases contractile function, eventually leading to cardiac dysfunction and dictating clinical outcomes in cardiac hypertrophy and heart failure [67]. These proteins were not measured in other models of increased activin A expression; however, we previously found that the reduced uterine perfusion pressure (RUPP) rat model of PE, which develops cardiac dysfunction following 5 days of placental ischemia-induced hypertension [68], also exhibits decreased cardiac α-MHC/β-MHC mRNA ratios [52]. The activin A-infused rats in the current study did not have elevated blood pressure, suggesting that activation of the fetal gene program is a direct effect of excess activin A on the heart.

Cardiac dysfunction was proposed as a state of reduced effectiveness of the natriuretic peptide system [69]. ANP and BNP, which are synthetized and secreted by cardiac myocytes, promote cardiovascular protection by antagonizing the actions of the renin–angiotensin–aldosterone system concerning blood pressure regulation and salt-water balance. The identification of these natriuretic peptides as sensitive markers of the cardiac load has profoundly impacted research into cardiovascular homeostasis and disease [70]. In the case of cardiac dysfunction, particularly in heart failure, despite dramatic increases in circulating levels of ANP and BNP, their effects become blunted [69]. Still, they serve as accurate diagnosis and prognosis markers for various cardiac disorders [58]. Increased circulating levels of ANP and BNP were described in PE patients, including those with reported impaired GLS [15,71,72,73]. Likewise, we and others found elevated ANP and BNP levels in the circulation and/or heart tissue of RUPP rats [52,74,75]. Previous studies demonstrated that activin A overexpression increases, whereas inhibition of ActIIR signaling following pressure overload or myocardial infarction decreases cardiac gene expression of both ANP and BNP in mice [62,63]. We were unable to detect a significant rise in either serum or heart levels of ANP and BNP. However, we showed that serum activin A is positively correlated to serum BNP. Since these earlier studies have measured ANP and BNP mRNA levels after 8–10 weeks of insult/treatment, the lack of effect of activin A on triggering marked natriuretic peptide release in the current study might be a consequence of the short duration of activin A infusion.

### 4.4. Study Limitations

Our study has some limitations that should be noted. For instance, continuous activin A infusion induced subclinical cardiac dysfunction in pregnant rats, similarly to what was reported in PE patients [9,10,12,14]. Whether this insult is sufficient to increase the risk for long-term cardiac disorders after PE is still unknown. As placental anti-angiogenic factors were also associated with cardiac dysfunction in PE, follow-up studies should examine whether an infusion of activin A with or without sFlt-1 and/or sEng into pregnant animals results in worsened GLS during and after gestation. Furthermore, future studies should determine whether placental and heart expression of activin A and ActIIR is increased in models of PE-induced cardiac dysfunction, such as in RUPP rats [52,68,74,75] and whether treatment with inhibitors of the activin A-ActIIR pathway improves GLS.

## 5. Conclusions

In conclusion, we found that circulating activin A in excess induces cardiac stress and subclinical dysfunction in pregnant rats, represented by increased heart β-MHC expression and decreased GLS. As the placenta is the major source of activin A during pregnancy and placental activin expression was positively correlated with circulating activin A levels in PE patients [29,31,37,41], our data provide evidence that activin A may serve as a causal link between placental abnormalities and cardiac dysfunction in PE.

## Figures and Tables

**Figure 1 cells-11-00742-f001:**
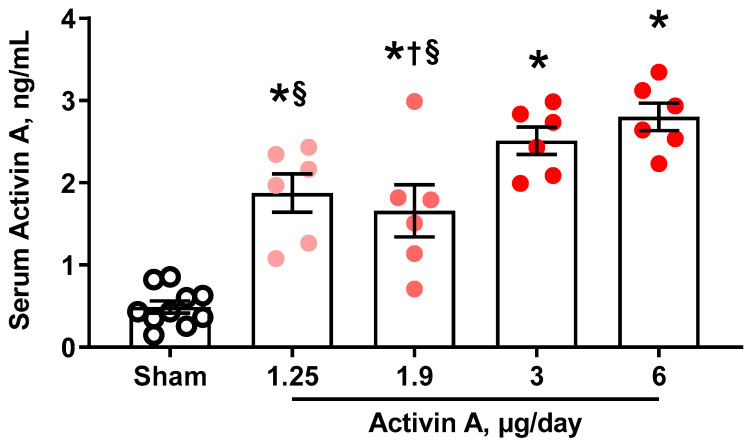
Circulating levels of activin A on gestational day 19 in sham (black open circles) and pregnant rats infused with 1.25 µg/day, 1.9 µg/day, 3 µg/day, and 6 µg/day of human recombinant activin A (solid red circles). Data are mean ± SEM. * *p* < 0.05 vs. sham, † *p* < 0.05 vs. 3 µg/day, § *p* < 0.05 vs. 6 µg/day.

**Figure 2 cells-11-00742-f002:**
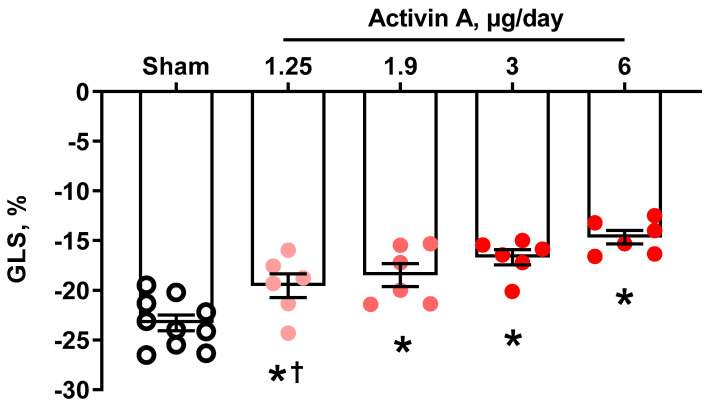
Cardiac global longitudinal strain (GLS) on gestational day 19 in sham (open black circles) and pregnant rats infused with 1.25 µg/day, 1.9 µg/day, 3 µg/day, and 6 µg/day of human recombinant activin A (solid red circles). Data are mean ± SEM. * *p* < 0.05 vs. sham, † *p* < 0.05 vs. 6 µg/day.

**Figure 3 cells-11-00742-f003:**
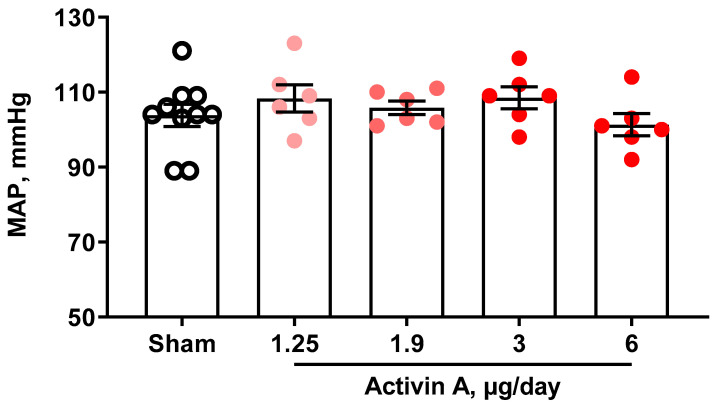
Mean arterial pressure (MAP) on gestational day 19 in sham (open black circles) and pregnant rats infused with 1.25 µg/day, 1.9 µg/day, 3 µg/day, and 6 µg/day of human recombinant activin A (solid red circles). Data are mean ± SEM.

**Figure 4 cells-11-00742-f004:**
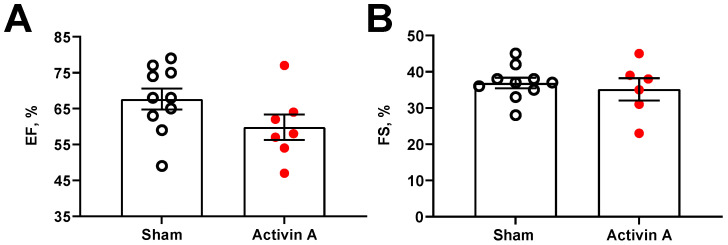
Cardiac parameters. Ejection fraction (EF; (**A**)); and fractional shortening (FS; (**B**)); measured on gestational day 19 in sham (open black circles) and activin A-infused (6 µg/day; solid red circles) pregnant rats. Data are mean ± SEM.

**Figure 5 cells-11-00742-f005:**
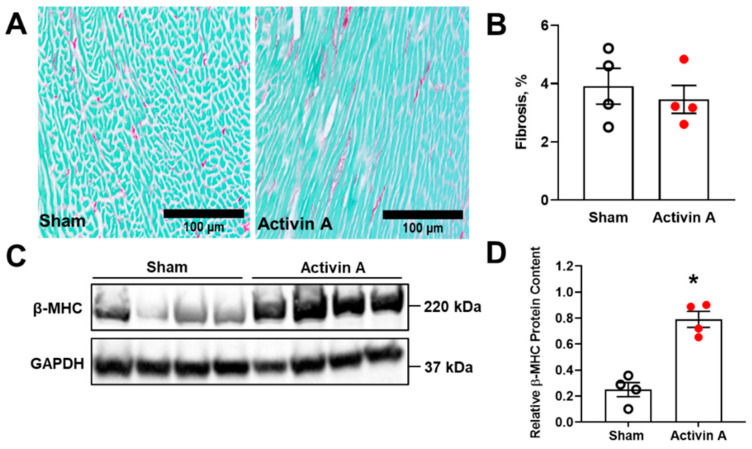
Markers of cardiac injury. Left ventricular fibrosis (**A**,**B**) indicated by Sirius Red and Fast Green staining, and beta-cardiac myosin heavy chain (β-MHC) protein content (**C**,**D**) on gestational day 19 in sham (open black circles) and activin A-infused (6 µg/day; solid red circles) pregnant rats. Data are mean ± SEM, * *p* < 0.05 vs. sham.

**Figure 6 cells-11-00742-f006:**
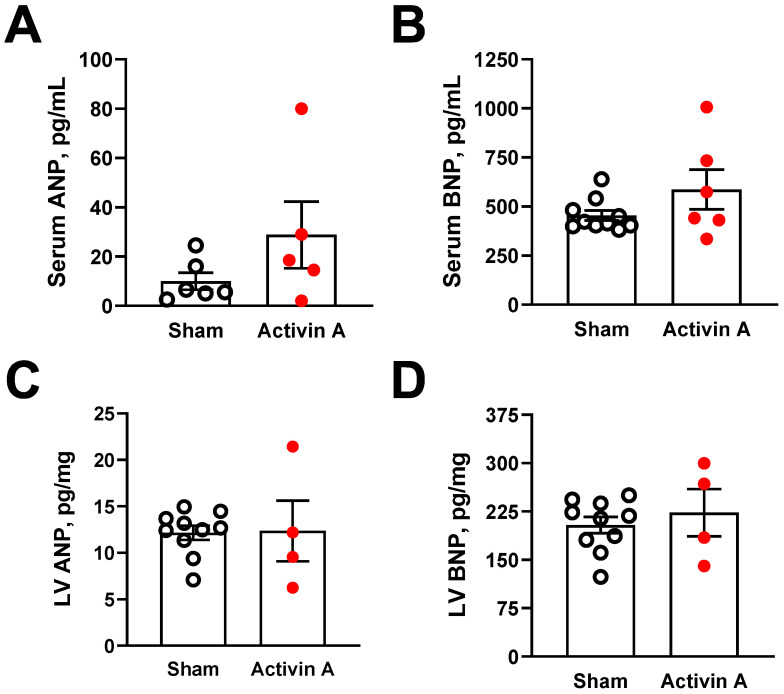
Circulating markers of cardiac injury. Serum (**A**,**B**) and left ventricular (LV; (**C**,**D**)); concentration of atrial natriuretic peptide (ANP) and brain natriuretic peptide (BNP) on gestational day 19 in sham (open black circles) and activin A-infused (6 µg/day; solid red circles) pregnant rats. Data are mean ± SEM.

**Figure 7 cells-11-00742-f007:**
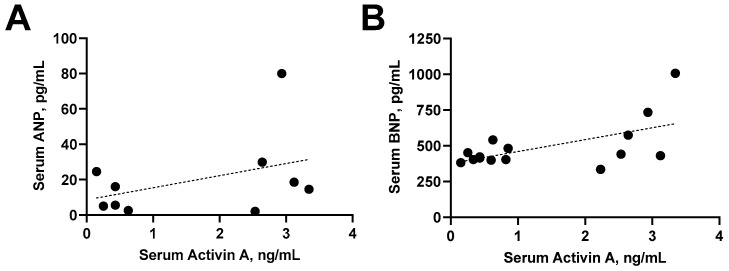
Relationship between activin A and circulating markers of cardiac injury. Correlation of serum activin A with serum atrial natriuretic peptide (ANP; r = 0.40, *p* = 0.25; (**A**)) or serum brain natriuretic peptide (BNP; r = 0.58, *p* = 0.02; (**B**)) in sham and activin A-infused (6 µg/day) pregnant rats.

**Table 1 cells-11-00742-t001:** Characteristics of sham vs. activin A (6 µg/day) infused rats on gestational day 19.

	Sham	Activin A	*p*-Value
n	10	6	
Body weight, g	302 ± 10	305 ± 8	0.85
Fetal weight, g	2.66 ± 0.124	2.74 ± 0.072	0.65
Placental weight, g	0.65 ± 0.0213	0.64 ± 0.026	0.89
Heart weight, g	0.84 ± 0.011	0.87 ± 0.064	0.24
Heart rate, bpm	405 ± 14	408 ± 11	0.91
Cardiac output, mL/min	87 ± 5	101 ± 5	0.08

## Data Availability

The authors confirm that data supporting the findings of this study are presented within the article. Raw data are available from the corresponding author upon reasonable request.

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
