# Peer review of "Sustained Elevated Circulating Activin A Impairs Global Longitudinal Strain in Pregnant Rats: A Potential Mechanism for Preeclampsia-Related Cardiac Dysfunction"

_cells, 2022, doi:10.3390/cells11040742_

Round 1

Reviewer 1 Report

This is a very well written manuscript based on interesting results elucidating the role of systemic Activin A infusion in pregnant rats and its association with cardiac function. It certainly would be interested, as the authors suggest in conclusion, to have administered Activin A in the RUPP rat model of preeclampsia. The link between Activin A and preeclampsia is tenuous in this study, however it is based on extensive published data that increased systemic concentration of Activin A was demonstrated in women with and following preeclampsia in conjunction with cardiac dysfunction. This is important work, which aims to understand the Activin A mechanisms on cardiac dysfunction in pregnancy potentially relevant to preeclampsia. Overall the results are interesting, and well interpreted in the context of the literature. I do, however, have a few comments that I think would help the readership of this manuscript.

  • The authors state that the main source of Activin A is placenta. Have they measured the expression of Activin A in placenta following systemic administration of the recombinant Activin A?
  • Given the differences observed in Lim et al [51]  with this study in terms of measures of PE-like characteristics what is the rational for choosing the doses 1.25, 1.9, 3 and 6 μg/day? This is over 10-fold less than what was previously used in vivo and demonstrated preeclampsia-like symptoms.
  • Was any diastolic dysfunction or LV hypertrophy observed?
  • No differences were observed in fetal and placental weight between the groups, however was there a differences in the resorption rate?
  • Another limitation was small n numbers in Figure 5 & Fig 6 C/D. Also, in Figure 6 C/D was LV concentration of ANP and BNP adjusted to total mg of protein?
  • Would be good to discuss potential cardiac-specific mechanism of Activin A given that in this study it did not lead to cardiac fibrosis and it did not affect MAP, BNP and ANP. What drives Activin A increase in preeclampsia?

Author Response

We thank the reviewer for their constructive feedback to improve our manuscript (cells-1585796), “Sustained elevated circulating Activin A impairs global longitudinal strain in pregnant rats: a potential mechanism for preeclampsia-related cardiac dysfunction.” Our response to each comment can be found below.

Response to Reviewer 1

This is a very well written manuscript based on interesting results elucidating the role of systemic Activin A infusion in pregnant rats and its association with cardiac function. It certainly would be interested, as the authors suggest in conclusion, to have administered Activin A in the RUPP rat model of preeclampsia. The link between Activin A and preeclampsia is tenuous in this study, however it is based on extensive published data that increased systemic concentration of Activin A was demonstrated in women with and following preeclampsia in conjunction with cardiac dysfunction. This is important work, which aims to understand the Activin A mechanisms on cardiac dysfunction in pregnancy potentially relevant to preeclampsia. Overall, the results are interesting, and well interpreted in the context of the literature. I do, however, have a few comments that I think would help the readership of this manuscript.

Comment 1: The authors state that the main source of Activin A is placenta. Have they measured the expression of Activin A in placenta following systemic administration of the recombinant Activin A?

Response 1: We did not measure Activin A in the placenta of these animals. It is thought that Activin A is mainly produced by the placenta in preeclampsia. However, for this study, Activin A was infused intraperitoneally into normal healthy pregnant rats; therefore, we did not expect to see increases in placental Activin A. We are currently conducting follow up studies to measure Activin A in the circulation, placental, and cardiac tissues of RUPP rats, and then correlate Activin A levels with echocardiogram parameters of cardiac systolic and diastolic function in this animal model of preeclampsia.

Comment 2: Given the differences observed in Lim et al [51] with this study in terms of measures of PE-like characteristics what is the rational for choosing the doses 1.25, 1.9, 3 and 6 μg/day? This is over 10-fold less than what was previously used in vivo and demonstrated preeclampsia-like symptoms.

Response 2: A further explanation of the selected doses has been added to the Materials and Methods section on lines 98-100, Results section on lines 163-165, and Discussion section on lines 241-245 of the revised manuscript. Our clinical data show that circulating Activin A levels are five times higher in preeclampsia compared to healthy pregnancies. These doses were selected to achieve this, and we found that 6 μg/day resulted in a five-fold increase in circulating Activin A levels of pregnant rats. To determine the direct effect of Activin A on the heart, independent of hypertension, we were not seeking a preeclampsia-like phenotype for this study. As mentioned in Response 1, our undergoing studies will explore the relationship between placental ischemia-induced hypertension, Activin A and cardiac dysfunction in the RUPP rat model.

Comment 3: Was any diastolic dysfunction or LV hypertrophy observed?

Response 3: We did not evaluate diastolic dysfunction in this batch of animals; however, we have preliminary data in few sham pregnant rats and pregnant rats infused with 6 μg/day Activin A, and we do not see any diastolic dysfunction: MV E, Sham, 774 ± 45 mm/s vs. Activin A, 715 ± 53 mm/s, p=0.45; MV A, Sham, 630 ± 46 mm/s vs. Activin A, 586 ± 43 mm/s, p=0.45; and MV E/A, Sham, 1.25 ± 0.04 vs. Activin A, 1.23 ± 0.09, p=0.85 (mean±SEM). In addition, we have not specifically measured left ventricular anterior/posterior wall thickness; however, as depicted in Table 1, we did not see a significant difference in whole heart weight.

Comment 4: No differences were observed in fetal and placental weight between the groups, however was there a difference in the resorption rate?

Response 4: Reabsorption rate was low across all the groups, and we did not see any significant differences: Sham, 2±1.3%; 1.25 μg/day, 4.5±2.0%; 1.9 μg/day, 4±1.8%; 3 μg/day, 0%; and 6 μg/day, 1.4±1.4% (mean±SEM).

Comment 5: Another limitation was small n numbers in Figure 5 & Fig 6 C/D. Also, in Figure 6 C/D was LV concentration of ANP and BNP adjusted to total mg of protein?

Response 5: Although sample sizes are small in Figure 5, we were able to detect a pronounced increase in β-MHC protein expression in heart homogenates, but no trends in fibrosis content in heart histological slides when comparing 6 ug/day Activin A-infused pregnant rats versus sham pregnant rats. On the other hand, ANP levels in many samples, especially those of the sham group were under the detection limit of the ELISA standard curve. Unfortunately, we had no remaining heart samples from 2 of the Activin A-infused pregnant rats to run the ANP and BNP assays. Despite these limitations, we were able to find a positive correlation between serum Activin A levels and serum BNP levels. Thus, we believe that increasing sample sizes would not change results and the overall conclusion of our study. We have provided information about this issue of reduced sample sizes in some experiments for clarification. Please see lines 222-227 in the Results section of the revised manuscript.

After homogenization of heart tissue samples, we have quantified total protein concentration in supernatants by a BCA assay. In addition to loading an equal amount of protein per lane of SDS-PAGE gels, data on β-MHC protein expression illustrated on Panels C and D of Figure 5 were normalized to GAPDH protein expression as assessed by Western Blotting. For data depicted on Panels C and D of Figure 6, an equal volume was loaded per well of ELISA kits, and ANP and BNP levels were normalized to total protein concentration in each sample. Please see lines 125-126 and 147-149 in the Materials and Methods section of the revised manuscript.

Comment 6: Would be good to discuss potential cardiac-specific mechanism of Activin A given that in this study it did not lead to cardiac fibrosis and it did not affect MAP, BNP and ANP. What drives Activin A increase in preeclampsia?

Response 6: Although we were unable to detect a statically significant rise in either serum or heart levels of ANP and BNP, increased serum Activin A was positively correlated to serum BNP in our pregnant rats. As mentioned in the Discussion section (lines 360-363), the lack of effect of Activin A on triggering a marked natriuretic peptide release may be a consequence of the short duration of Activin A infusion. Although, it might also be due to a limitation of the ELISA kits employed to measure ANP and BNP in the current study, as described in Response 5. Indeed, we did detect a cardiac-specific mechanism of injury where Activin A-infused pregnant rats had a shift in MHC isoforms. Culminating evidence indicates that increased β-MHC expression decreases contractile function, eventually leading to cardiac dysfunction and dictating clinical outcomes in cardiac hypertrophy and heart failure [67]. We have reworded the relevant section in the Discussion section of the revised manuscript to clarify this issue (lines 335-337).

As mentioned in the Introduction section (lines 63-67), Activin A is a normal component of the placenta and other reproductive organs, playing a tightly regulated role in embryo development and pregnancy. Increased levels of Activin A have been reported in blood and placental tissue of patients with preeclampsia compared to both patients with gestational hypertension and normal pregnant women. In preeclampsia, placental malperfusion stimulates the release of placental factors into the maternal circulation, suggesting increased circulating levels of Activin A in preeclampsia is derived from exacerbated placental release of Activin A. Indeed, placental Activin A expression has been positively correlated with circulating Activin A levels in preeclamptic patients [29, 31, 37, 41]. However, there are in vitro studies showing that trophoblasts as well as endothelial and immune cells secret Activin A when exposed to pro-inflammatory and stress oxidative conditions, both hallmarks of preeclampsia [23]. Therefore, although the exact mechanism is unclear at this time, emerging evidence suggest both the placenta and other cell type may be responsible for elevated Activin A in preeclampsia. A further explanation of the possible sources for Activin A in preeclampsia has been added to the Discussion section on lines 266-272 of the revised manuscript.

Reviewer 2 Report

The manuscript by Brakrania et al., point a very interesting issue relating preeclampsia with cardiac dysfunction. The manuscript is well designed with a comprehensive introduction to a complex topic. The methods are accurate to the aims of the work and well described. The results are well described and the explanation for the use of different parameters to determine fibrosis are appropriate. The discussion is complete and well references.
I have some recommendations:
1.    The title beginning with the word excess is not clear maybe “High levels of …”
2.    An explanation about the use of the higher dose of Activin A should be do it.
3.    The paragraph in the page 7 explains the results of ANP and BNP in the circulation and tissue. However is hard to understand the explanation from line 213 to 220: “ Although there were no statistically significant differences in either serum or left ventricular ANP levels 214 between groups (Fig 6A, 6C), 6 out 12 serum ANP values in the Sham group were undetectable by the ELISA kit compared with 1 out 6 values in the Activin A-infused group. Likewise, circulating and left ventricular BNP levels were statistically comparable bettween groups (Fig 6B, 6D), but only 1 out 12 serum BNP value was undetectable by the 218 ELISA kit in the Sham group. Still, circulating Activin A was positively correlated to circulating BNP, suggesting that Activin A induces cardiac injury (Fig 7A, 7B)”. It should be re-written.
4.    It is worthy to inform of the limitations of the study. A header should be used before the line 346.

Author Response

We thank the reviewer for their constructive feedback to improve our manuscript (cells-1585796), “Sustained elevated circulating Activin A impairs global longitudinal strain in pregnant rats: a potential mechanism for preeclampsia-related cardiac dysfunction.” Our response to each comment can be found below.

Response to Reviewer 2

The manuscript by Bakrania et al., point a very interesting issue relating preeclampsia with cardiac dysfunction. The manuscript is well designed with a comprehensive introduction to a complex topic. The methods are accurate to the aims of the work and well described. The results are well described and the explanation for the use of different parameters to determine fibrosis are appropriate. The discussion is complete and well references. I have some recommendations:

Comment 1: The title beginning with the word excess is not clear maybe “High levels of …”

Response 1: We have changed the title as following: “Sustained elevated circulating Activin A impairs global longitudinal strain in pregnant rats: a potential mechanism for preeclampsia-related cardiac dysfunction.”

Comment 2: An explanation about the use of the higher dose of Activin A should be do it.

Response 2: A further explanation of the selected doses has been added to the Materials and Methods section on lines 98-100, Results section on lines 163-165, and Discussion section on lines 241-245 of the revised manuscript. Our clinical data show that circulating Activin A levels are five times higher in preeclampsia compared to healthy pregnancies. Therefore, we trialed four doses to achieve this in pregnant rats. We found that 5 day-intraperitoneal infusion of Activin A at 6 μg/day resulted in a five-fold increase in circulating Activin A levels compared to the Sham group. Thus, all the biochemical experiments were conducted comparing the 6 ug/day Activin A-infused pregnant rats versus the Sham pregnant rats.

Comment 3: The paragraph in the page 7 explains the results of ANP and BNP in the circulation and tissue. However is hard to understand the explanation from line 213 to 220: “Although there were no statistically significant differences in either serum or left ventricular ANP levels between groups (Fig 6A, 6C), 6 out 12 serum ANP values in the Sham group were undetectable by the ELISA kit compared with 1 out 6 values in the Activin A-infused group. Likewise, circulating and left ventricular BNP levels were statistically comparable between groups (Fig 6B, 6D), but only 1 out 12 serum BNP value was undetectable by the ELISA kit in the Sham group. Still, circulating Activin A was positively correlated to circulating BNP, suggesting that Activin A induces cardiac injury (Fig 7A, 7B)”. It should be re-written.

Response 3: We have rewritten this paragraph for clarification and to improve readability. Please see lines 220-229 in the Results section of the revised manuscript.

Comment 4: It is worthy to inform of the limitations of the study. A header should be used before the line 346.

Response 4: We have added a heading above line 366 in the revised manuscript for easier identification of the study limitations paragraph.